# Molecular Characterization of the Infectious Laryngotracheitis Virus (ILTV) Involved in Poultry Outbreaks Reveals the Virus Origin and Estimated Spreading Route

**DOI:** 10.3390/v17020213

**Published:** 2025-01-31

**Authors:** Jorge Luis Chacón, Ruy D. Chacón, Henrique Lage Hagemann, Claudete S. Astolfi-Ferreira, Cesar Nunes, Luiz Sesti, Branko Alva, Antonio J. Piantino Ferreira

**Affiliations:** 1CEVA Animal Health, Rua Manoel Joaquim Filho, 303, São Paulo 13148-115, Brazil; jorge.chacon@ceva.com (J.L.C.); cesar.nunes@ceva.com (C.N.); luiz.sesti@ceva.com (L.S.); branko.alva@ceva.com (B.A.); 2Department of Pathology, School of Veterinary Medicine, University of São Paulo, Av. Prof. Orlando Marques de Paiva, 87, São Paulo 05508-900, Brazil; ruychaconv@alumni.usp.br (R.D.C.); henrique.trick@alumni.usp.br (H.L.H.); csastolfi@gmail.com (C.S.A.-F.)

**Keywords:** chicken, infectious laryngotracheitis virus, *Gallid alphaherpesvirus 1*, molecular characterization, infected cell protein 4

## Abstract

Infectious laryngotracheitis outbreaks have been observed in a short period of time in broiler, layer, and broiler breeder flocks, resulting in clinical signs and high mortality. The affected farms are located in the same geographical area, which is a high-density poultry region of Brazil. To estimate the potential origin of the virus or viruses that caused the outbreaks, the ILTVs detected at six companies were molecularly characterized by sequencing two fragments of the ICP4 gene and then compared with previous field and vaccine viruses detected in the country. The sequencing results revealed that all farms investigated were infected with a nonvaccine-origin virus. Phylogenetic analysis revealed that all farms were infected by the same virus classified as genotype VI. In addition, the ILTV detected in the present study was compared with that of viruses previously detected in egg-layer poultry regions in the country. The viruses detected in the recent outbreaks were indistinguishable, with one of them (VI-4) suggesting a possible route of transmission. This study describes for the first time severe ILT outbreaks in meat-type poultry in Brazil that spread quickly, and the phylogenetic analysis suggests the potential origin of the virus and route of transmission.

## 1. Introduction

Infectious laryngotracheitis (ILT) is a highly contagious upper respiratory disease of poultry that causes great economic loss because of increased mortality, decreased growth rates, and reduced egg production associated with infection [1].

ILT is caused by infectious laryngotracheitis virus (ILTV), a herpesvirus of the family *Orthoherpesviridae*, classified as *Iltovirus gallidalpha1* (previously classified as *Gallid alphaherpesvirus 1*) [2]. The virion presents an icosahedral nucleocapsid surrounded by a protein tegument layer encapsulated by an external envelope containing glycoproteins [1]. The ILTV genome is a linear double-stranded DNA type of approximately 153 kilobases. The genome is composed of unique long (UL) and unique short (US) regions flanked by inverted repeats [3].

Antigenically, ILTV is conserved, and molecular characterization based on complete genomes allows for its classification into IX genotypes [4]. However, molecular classification based on PCR and sequencing of the infected cell protein 4 (ICP4) gene are still the most widely used methods to genotype ILTV strains worldwide [4,5,6,7,8].

Mild and severe clinical forms of the disease are characterized by high mortality, severe dyspnea, bloody mucus expectoration, and suffocation in severe cases. Sources of ILTV include birds with acute or latent infection and secretion-contaminated materials such as dust, litter, and fomites [1,9].

Currently, disease control is based on vaccination with either live attenuated vaccines or recombinant viral vector vaccines [10]. Live attenuated vaccines include (1) vaccines attenuated by multiple passages in embryonated eggs or chicken embryo origin (CEO) and (2) a vaccine generated by multiple passages in tissue culture or tissue culture origin (TCO). In many countries, these vaccines have been forbidden because they can regain virulence during consecutive passages in chickens [9]. In addition, whole-genome sequence analysis revealed that ILTV live attenuated vaccines are composed of mixed viral subpopulations, and it is suspected that the selection of virulent CEO vaccine subpopulations may lead to the emergence of outbreak-related viruses [11]. Recombinant viral vector vaccines confer no risk of reversion of virulence, efficacious protection against clinical disease, and somewhat reduced efficacy against ILTV replication and excretion [12].

ILT outbreaks were described previously in commercial egg-layer areas of two states in Brazil. Initially, the TCO and CEO vaccines were used for control of the first outbreaks [13]. Currently, because of the safety risks associated with live attenuated vaccines, only recombinant viral vector vaccines are allowed for vaccination in Brazil. Live vaccine-related viruses and field strains can cause ILT outbreaks [7,13,14,15]. Previous publications have shown the circulation of CEO and TCO vaccine viruses and field strains in vaccinated and nonvaccinated regions of Brazil [16,17].

At the beginning of the second half of 2024, severe ILT outbreaks were observed in a high-density poultry state where ILT had never been diagnosed and therefore vaccination was never carried out. Many ILT cases have been described in a short period of time in broiler and broiler breeder farms, leading to an increase in mortality and a reduction in fertile egg production.

The objectives of this study were to genetically characterize the ILTV detected in outbreaks to estimate the origin of the virus and eventually assess possible transmission routes.

## 2. Materials and Methods

### 2.1. ILT Outbreaks and Clinical Samples

Respiratory signs of suspected ILT have been observed in broiler and broiler breeder flocks located in the northern and western regions of the state of Paraná since the middle of 2024. Clinical signs detected from 28 days of age in broiler flocks included conjunctivitis, swelling of the head, nasal discharge, watery/foamy eyes, severe dyspnea, and expectoration of bloody mucus. The presence of ILTV was confirmed by molecular detection, and notification was performed by the National Poultry Health authorities. The affected flocks were in a high-density poultry region, and all cases occurred within a relatively small area in which the distance between affected farms was less than 150 km. The mortality rates of broiler and broiler breeder flocks varied from 7 to 12.5% and 3.5 to 48%, respectively. The affected broiler breeder flocks presented a reduction in fertile egg production (12.3–20%), with recovery to normal parameters after four weeks from the beginning of the clinical signs. The affected farms had never been vaccinated against ILT since the infection had not been present in the region. Organs and tissues of dead birds in broiler and broiler breeder flocks were collected over a period of two months and sent to the Laboratory of Avian Diseases of the School of Veterinary Medicine and Animal Science of the University of São Paulo (USP), São Paulo, Brazil (Table 1). The samples consisted of a pool of organs from five birds from the same flock that included tracheas, lungs, trigeminal ganglia, and palpebrae.

### 2.2. DNA Extraction for ILTV Detection

The samples were macerated in phosphate-buffered saline (PBS) at pH 7.2 at a 1:1 volume ratio. The samples were subjected to three cycles of thermal shock (freezing at −80 °C for 10 min and thawing at 56 °C for 1 min) and 12,000× *g* for 20 min at 4 °C. Then, 200 μL of the supernatant was collected for nucleic acid extraction. DNA was extracted with the ReliaPrep™ Viral TNA Miniprep System Kit (Promega, Madison, WI, USA) according to the conditions recommended by the manufacturer, quantified with a NanoDrop One (Thermo Scientific^TM^, Wilmington, DE, USA) and stored at −20 °C until further use.

The detection of ILTV was carried out using a previously published quantitative polymerase chain reaction (qPCR) assay [18]. This reaction is based on the amplification of glycoprotein E (gE) using the primers Glic-EF: ACGCACATGCCCTCGAA and Glic-ER: GGTCCGGGACTGCCAATTA and the PowerUp™ SYBR^®^ Green Master Mix (Applied Biosystems, Austin, TX, USA) in a QuantStudio3 Real Time PCR System (Applied Biosystems, Marsiling, Singapore).

### 2.3. Sequencing and Phylogenetic Analysis

Molecular characterization was performed via the amplification and sequencing of two fragments of the ICP4 gene. These fragments were amplified using the primers ICP4-1F: ACTGATAGCTTTTCGTACAGCACG and ICP4-1R: CATCGGGACATTCTCCAGGTAGCA (687 bp fragment), the primers ICP4-2F: CTTCAGACTCCAGCTCATCTG and ICP4-2R: AGTCATGCGTCTATGGCGTTGAC (631 bp fragment) and PCR conditions, according to Chacón and Ferreira [19].

The PCR products were extracted from a 1.5% agarose gel using the Wizard^®^ SV Gel and PCR Clean-Up System Kit (Promega, Madison, WI, USA). Dideoxynucleotide sequencing was performed on a 3500xL Genetic Analyzer using the BigDye^TM^ Terminator v3.1 Cycle Sequencing Kit (Applied Biosystems, Carlsbad, CA, USA). The sequencing products were assembled with Geneious Prime^®^ version 2020.0.7 and aligned with Clustal Omega version 1.2.2. [20], along with other public ILTV sequences retrieved from GenBank.

For the phylogenetic analysis, PhyML version 3.0 [21] was used to construct a maximum likelihood tree via the Bayes branch support method and the nucleotide substitution model HKY85 + I, which was estimated via SMS software [22]. The phylogenetic tree obtained was edited with the iTOL version 6 program [23]. The sequences generated in this study were deposited in GenBank under accession numbers PQ658695-PQ658712.

## 3. Results

### 3.1. Gross Findings

The severity and stage of the macroscopic lesions located in the upper respiratory tract varied. The conjunctival lesions included swelling of the eyelids and conjunctival edema (Appendix A). Edema and congestion in the infraorbital sinuses were also detected. Congestion and inflammation were observed in the laryngeal and tracheal mucosa. Mucous exudate, blood casts, and yellowish caseous plaques were present inside the trachea and larynx with partial or total obstruction of the tracheal lumen.

### 3.2. Phylogenetic Analysis

The phylogenetic analysis was carried out considering the nine previously defined genotypes [4,24]. In this way, the sequences generated in this study were all grouped within genotype VI (Figure 1). This genotype is distributed in different countries, and most sequences were previously published in Brazil between 2002 and 2020 [5,17,19,25]. Within genotype VI, five subclusters can be identified. Subcluster VI-4 included all the sequences in the present study and Brazilian sequences originating from Santa Catarina in 2020. In addition, other sequences have been isolated from America, Peru (2014), Canada (2019–2021), the USA (2004–2006), and western Asia, including Turkey (2015–2020) and Iran (2019). Other Brazilian sequences belonging to cluster VI correspond to previous outbreaks between 2002 and 2020, including those emerging in 2002 in Bastos, SP (subcluster VI-1), in 2009 in Guatapará, SP (subcluster VI-2), and in 2010 in Itanhandu, MG (subcluster VI-3).

### 3.3. Nucleotide Polymorphisms and Genetic Profiles of ILTV Genotypes

A comparison of the nucleotide polymorphisms in the two sequenced regions of the ICP4 gene was also carried out to compare the patterns with those of the reference sequence SA2 (accession number: NC_075683.1) and establish the genetic profile of each genotype (Table 2). A total of 50% (9 / 18) of the polymorphic sites in nucleotides constitute changes in the amino acid sequence. In general, the polymorphism profiles of the genotypes allow for them to be differentiated. An exception occurred in the case of the reference strain SA2, which presented a profile corresponding to genotypes VII, VII, and IX. Similarly, genotypes I, II, and III presented similar profiles related to the TCO vaccine prototype. Genotypes IV and V related to the CEO vaccine prototype also presented similar profiles, with characteristic deletions at positions 262 and 273. All the sequences related to the outbreaks in this study presented identical profiles, matching each other as well as the 2020 sequences from Santa Catarina. This suggests that they may have a common origin. This genotype is called VI-4, and the prototype was named São Ludgero, after the city where the oldest strain of this genotype was found in Brazil. The polymorphism profiles included G438A, G456A, G598A, A795G, A811G, C3905T, T3957C, and T4298C. The oldest prototypes of the other field strain genotypes found in Brazil (VI-1, VI-2, and VI-3) also differed significantly on the basis of their city of origin.

## 4. Discussion

Since the middle of 2024, ILT outbreaks have been reported in a densely populated poultry area in Paraná State, Brazil. The disease spread rapidly, impacting broiler and broiler breeder flocks from six large companies. Genetic studies were conducted to characterize the ILTV detected in the state of Paraná and compare it with previously detected viruses in the country to understand its origin and determine potential routes of transmission.

The observed clinical signs and reduction in productivity reported in these outbreaks are quite similar to those reported in other outbreaks caused by highly virulent strains [1,26]. Severe cases of the disease observed during outbreaks can be attributed to the strength of the virus strain, the lack of immunity against ILTV in unvaccinated flocks, and coinfection with other pathogens, such as variant strains of the infectious bronchitis virus found in the area [27,28].

The viruses found at the six companies were genetically indistinguishable, suggesting that all the flocks were contaminated by the same virus. The flocks from all the companies are located within a maximum distance of 150 km from each other. This finding indicates that the biosecurity measures in place at the affected companies were not effective in preventing ILTV contamination. The farms are located in a densely populated poultry area, which makes it easier for the virus to spread. Additionally, the role of wind in spreading the virus is likely an important variable, as has been described elsewhere [29].

Previous studies of whole-genome sequences have classified ILTV into six genotypes, with genotype VI containing nonvaccine viruses [4]. Phylogenetic analysis of the ICP4 gene sequences revealed that the viruses involved in recent outbreaks belong to genotype VI. However, viruses within this genotype can be further differentiated and grouped into subclusters. The analysis indicates that four different field strains are currently circulating in Brazil. These strains were isolated from outbreaks in egg-producing regions: Bastos in the state of São Paulo in 2002 [19], Guatapará in São Paulo in 2009 [17], Itanhandú in the state of Minas Gerais in 2010 [25], and the state of Santa Catarina in 2020 [30]. To aid in future research, we propose naming these subclusters of viruses VI-1, VI-2, VI-3, and VI-4.

Transport of infected birds or contaminated litter is the main route of infection spread to uninfected areas [1,31]. The virus found in recent outbreaks was first identified four years ago in a poultry area over 800 km away [30]. The first detection of the VI-BRA4 strain in eastern Santa Catarina and its recent appearance in western Paraná suggest that the virus may have spread through the transportation of infected birds or contaminated materials, such as litter.

In 2020, outbreaks in Santa Catarina were quickly controlled through mandatory vaccination with at least two doses of recombinant vector vaccines. The detection of the virus responsible for the 2024 outbreaks four years later highlights the challenge of eliminating ILTV. This finding shows that while biosecurity measures, including vaccination, are essential for controlling the disease, they do not completely eliminate the presence or spread of ILTV [1]. Furthermore, eliminating ILTV in multiage farms is complicated by the virus’s ability to establish latent infections [32]. Previous studies on ILTV in Brazil reported the presence of CEO and TCO vaccine viruses, along with at least four different field strains. The goal of this study was to determine whether any of these previously identified viruses caused the 2024 outbreak. The results indicate that the molecular methods used for virus characterization effectively distinguished between the previous vaccine strains and the field-generated ILTV strains. Additional and comparative studies of complete genomes in the region are needed to elucidate other possible transmission routes of ILTV in Brazil in the global context.

## 5. Conclusions

The genetic analysis used in this study allowed for the identification of a rapidly spreading specific strain of ILTV in broiler and broiler breeder companies within a densely populated poultry region, as well as the virus’s origin.

Genetic analyses of the ILTV isolates implicated in the Paraná outbreaks identified them as members of genotype VI, a group comprising nonvaccine viruses. Phylogenetic analysis further subdivided this genotype into four distinct subclusters, with the outbreak virus belonging to a lineage first identified in Santa Catarina in 2020.

This study underscores the importance of molecular characterization for differentiating between vaccine-derived and field-derived ILTV strains. The proposed nomenclature for Brazilian ILTV strains (VI-1 to VI-4) establishes a framework for monitoring viral evolution and epidemiology within the region.

To prevent future outbreaks, enhanced biosecurity measures, strategic vaccination protocols, and stringent control of poultry transport are crucial.

## Figures and Tables

**Figure 1 viruses-17-00213-f001:**
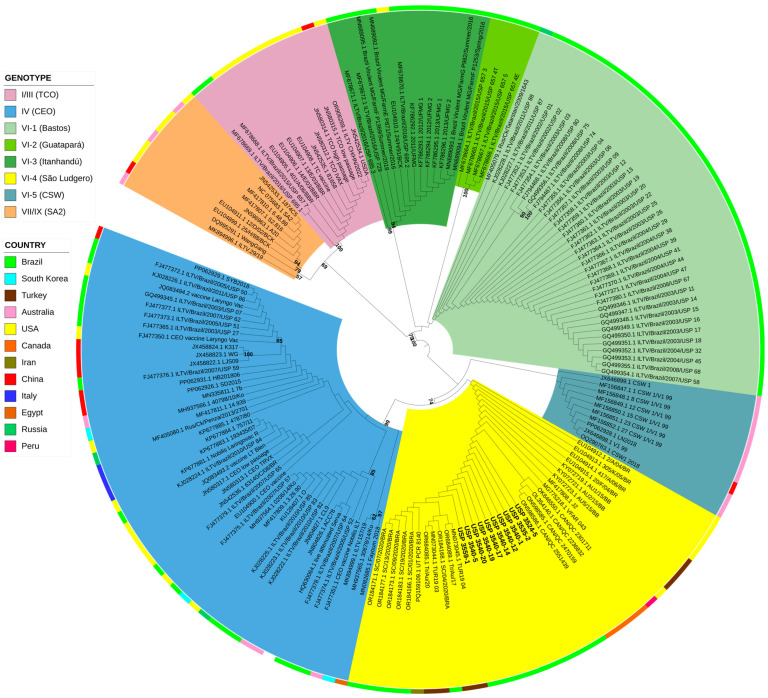
Maximum-likelihood phylogenetic tree of ILTV including 164 ICP4 sequences. Brazilian strains generated in this study are highlighted in bold. The nucleotide alignment was 1179 base pairs (bp) in size and consisted of two concatenated fragments of the ICP4 gene (630 and 549 bp in length). The tree was inferred under an HKY85 + I substitution model with the Bayes branch support method. The genotypes of the clusters are differentially colored, and the prototype is indicated in parentheses in the legend. The country of detection is annotated in the outer ring.

**Table 1 viruses-17-00213-t001:** Epidemiological and clinical information of the studied cases of ILT.

Company	Farm/Flock	ID	Bird	Age	Sample (s)
A	1	USP-3524-5	broiler	31 days	trachea
	2	USP-3524-6	broiler	40 days	trachea
	3	USP-3524-7	broiler	41 days	trachea
	4	USP-3524-8	broiler	40 days	trachea
	5	USP-3559-1	layer	12 weeks	trachea, palpebrae
B	7	USP-3540-1	broiler	N.I.	trachea, lung
C	8	USP-3540-5	broiler	N.I.	trachea, lung
	9	USP-3540-12	breeder	N.I.	trachea, lung
D	10	USP-3540-14	broiler	N.I.	trachea
	11	USP-3540-17	breeder	N.I.	trachea
E	12	USP-3540-19	broiler	N.I.	trachea
F	13	USP-3540-20	broiler	N.I.	trachea

N.I.: not informed.

**Table 2 viruses-17-00213-t002:** Genetic profiles of the ILTV genotypes considering the nucleotide polymorphisms of the ICP4 gene of the studied strains.

Strain	Country	Year	Genotype	Prototype	ICP4 Position																			
					262–273	438	456	539 ^†^	598 ^†^	606	622 ^†^	684	724–732	795	811 ^†^	3905 ^†^	3957	3981	4012 ^†^	4016 ^†^	4047	4298 ^†^	4339 ^†^	4377
NC_075683.1_SA2	Australia	1983	VII/IX		GCCCAAGACGCG	G	G	G	G	A	C	A	TCCTCTTCC	A	A	C	T	C	A	A	G	T	T	C
JN596963.1_A20	Australia	1966	VII/IX		.	.	.	.	.	.	.	.	.	.	.	.	.	.	.	.	.	.	.	.
MF417807.1_S2.816	USA	2002	VII/IX		.	.	.	.	.	.	.	.	.	.	.	.	.	.	.	.	.	.	.	.
JN542534.1_USDA_	USA	1960	I	**TCO**	.	.	A	.	.	.	.	.	.	G	G	.	.	T	G	G	.	C	A	A
JN580312.1_TCO_IVAX_	USA	1983	II		.	.	A	.	.	.	.	.	.	G	G	.	.	T	G	.	.	C	A	A
EU104905.1_401/A/06/BBR	USA	2006	III		.	.	A	.	.	.	.	.	.	G	G	.	.	T	G	.	.	C	A	A
JN580313.1_CEO_TRVX_	USA	1983	IV	**CEO**	―	A	A	.	A	.	.	.	.	G	G	T	C	.	.	.	A	C	.	A
JQ083494.2_vaccine_Laryngo_Vac	USA	1975	IV		―	A	A	.	A	.	.	G	.	G	G	T	C	.	.	.	A	C	.	A
JN542536.1_63140/C/08/BR	USA	2006	V		―	A	A	.	A	.	.	.	.	G	G	T	C	.	.	.	A	C	.	A
MF417811.1_14.939	USA	2014	V		―	A	A	.	A	.	.	.	.	G	G	T	C	.	.	.	A	C	.	A
MF417808.1_J2	USA	2008	VI-4		.	A	A	.	A	.	.	.	.	G	G	T	C	.	.	.	.	C	.	.
OR184166.1_SC/01/2020/BRA	Brazil—SC	2020	VI-4	**São Ludgero**	.	A	A	.	A	.	.	.	.	N	N	T	C	.	.	.	.	C	.	.
OR184168.1_SC/04/2020/BRA	Brazil—SC	2020	VI-4		.	A	A	.	A	.	.	.	.	N	N	T	C	.	.	.	.	C	.	.
OR184183.1_SC/19/2020/BRA	Brazil—SC	2020	VI-4		.	A	A	.	A	.	.	.	.	N	N	T	C	.	.	.	.	C	.	.
**USP-3524-5**	Brazil—PR	2024	VI-4		.	A	A	.	A	.	.	.	.	G	G	T	C	.	.	.	.	C	.	.
**USP-3535-2**	Brazil—SC	2024	VI-4		.	A	A	.	A	.	.	.	.	G	G	T	C	.	.	.	.	C	.	.
**USP-3540-1**	Brazil—PR	2024	VI-4		.	A	A	.	A	.	.	.	.	G	G	T	C	.	.	.	.	C	.	.
**USP-3540-5**	Brazil—PR	2024	VI-4		.	A	A	.	A	.	.	.	.	G	G	T	C	.	.	.	.	C	.	.
**USP-3540-12**	Brazil—PR	2024	VI-4		.	A	A	.	A	.	.	.	.	G	G	T	C	.	.	.	.	C	.	.
**USP-3540-14**	Brazil—PR	2024	VI-4		.	A	A	.	A	.	.	.	.	G	G	T	C	.	.	.	.	C	.	.
**USP-3540-17**	Brazil—PR	2024	VI-4		.	A	A	.	A	.	.	.	.	G	G	T	C	.	.	.	.	C	.	.
**USP-3540-19**	Brazil—PR	2024	VI-4		.	A	A	.	A	.	.	.	.	G	G	T	C	.	.	.	.	C	.	.
**USP-3540-20**	Brazil—PR	2024	VI-4		.	A	A	.	A	.	.	.	.	G	G	T	C	.	.	.	.	C	.	.
**USP-3559-1**	Brazil—PR	2024	VI-4		.	A	A	.	A	.	.	.	.	G	G	T	C	.	.	.	.	C	.	.
FJ477352.1_ILTV/Brazil/2002/USP-01	Brazil—SP	2002	VI-1	**Bastos**	.	.	.	.	.	.	.	.	.	G	G	.	C	.	.	.	.	C	.	.
GQ499354.1_ILTV/Brazil/2007/USP-58	Brazil—SP	2007	VI-1		.	.	.	.	.	.	.	.	.	G	G	.	C	.	.	.	.	C	.	.
KJ028228.1_ILTV/Brazil/2011/USP-88	Brazil—SP	2011	VI-1		.	.	.	.	.	.	.	.	.	G	G	.	C	.	.	.	.	C	.	.
MF678664.1_ILTV/Brazil/2015/USP-657-3	Brazil—SP	2015	VI-2	**Guatapará**	.	.	.	.	.	C	.	.	―	G	G	.	C	.	.	.	.	C	.	.
MF678667.1_ILTV/Brazil/2015/USP-657-5	Brazil—SP	2015	VI-2		.	.	.	.	.	C	.	.	―	G	G	.	C	.	.	.	.	C	.	.
MN689095.1_MG/FarmF-P1369/Summer/2018	Brazil—MG	2018	VI-2		.	.	.	T	.	.	T	.	―	G	N	.	C	.	.	.	.	C	.	.
KF786292.1_2011/UFMG	Brazil—MG	2011	VI-3	**Itanhandú**	.	.	.	T	.	.	T	.	.	G	.	.	C	.	.	.	.	C	.	.
KF786296.1_2013/UFMG-2	Brazil—MG	2013	VI-3		.	.	.	T	.	.	T	.	.	G	.	.	C	.	.	.	.	C	.	.
MF678670.1_ILTV/Brazil/2016/USP-695-2	Brazil—SP	2016	VI-3		.	.	.	T	.	.	T	.	.	G	.	.	C	.	.	.	.	C	.	.

Nucleotide substitutions: “―” indicates gap; “.” indicates conserved; colors indicate polymorphism; “^†^” indicates a change in amino acids. Nucleotide position according to the reference strain SA2 (accession number: NC_075683.1). The Brazilian strains used in this study are highlighted in bold.

## Data Availability

The sequences generated in this study were deposited in GenBank under accession numbers PQ658695-PQ658712.

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
