# Peer review of "Molecular Characterization of the Infectious Laryngotracheitis Virus (ILTV) Involved in Poultry Outbreaks Reveals the Virus Origin and Estimated Spreading Route"

_viruses, 2025, doi:10.3390/v17020213_

Round 1

Reviewer 1 Report

Comments and Suggestions for Authors

The authors present data from a study named „Molecular characterization of infectious laryngotracheitis virus (ILTV) involved in poultry outbreaks reveals virus origin and estimated spreading route“. The results show the anticoccidial effects of the different herbal and orthodox drugs. 

 Specific comments:

L 76 – 92 In the Materials and methods section, it would be useful to add a map of the farms where the virus was confirmed.

L 179 – 182 Authors should be add a reference for co-infections with infectious bronchitis virus.

The manuscript is very clear and the clinical cases are well described.

Author Response

Dear Editors and Reviewers:
We thank the reviewers for their thoughtful comments and constructive suggestions concerning our manuscript entitled “Molecular characterization of infectious laryngotracheitis virus (ILTV) involved in poultry outbreaks reveals virus origin and estimated spreading route” (viruses-3388251), which enabled us to resubmit a clearly improved manuscript. We highlighted the amendments in the revised manuscript, and responded, point by point to, the comments as listed below.

Reviewer #1: 
The authors present data from a study named „Molecular characterization of infectious laryngotracheitis virus (ILTV) involved in poultry outbreaks reveals virus origin and estimated spreading route“. The manuscript is very clear and the clinical cases are well described.
R0. We greatly appreciate for the comments on our manuscript.

Q1. L 76 – 92 In the Materials and methods section, it would be useful to add a map of the farms where the virus was confirmed.
R1. We thank the reviewer for this observation. Although the affected companies notified the outbreaks to National Poultry Health authorities, they prefer not to be identified. To present a map could reveal sensitive information. We apologize for this impossibility and hope for your understanding.

Q2. L 179 – 182 Authors should be add a reference for co-infections with infectious bronchitis virus.
R2. We thank the reviewer for this observation. We have added references on coinfection of ILTV and IBV as suggested. Changes are highlighted in yellow in the discussions and in the reference list.

Reviewer 2 Report

Comments and Suggestions for Authors

Comments to author:

The authors aimed to describe a case of ILTV in Broiler and broiler breeder flocks located in the Northern and Western regions of Parana State of Brazil in the second semester of 2024. Then, genetically characterized based on the amplification of the ICP4 gene in two fragments, and phylogenetically analysed. Finally, possible transmission routes discussed.

 General points:

Strength: Science is OK.

ILTV genotypes in broilers and breeder broilers in Brazil will add circulating genotypes in Brazil and would add information to the ILTV genotypes literature.

Weakness: One major issue is the lack of novelty of the work and limited interest in Brazil. Moreover, results analysis is just based on 164 ICP4 sequences, and other written results are descriptive. A complete genome analysis may give more information on phylogeny and the presence of any recombination events.

 Specific points:

1. Figure 1: The labeling of sequence names on the phylogenetic tree is inconsistent. For example, VI-1 is mentioned with the word “ILTV,” and for other sequence names, it is removed. May be just use the accession number with year and country, and specific names of the samples. In addition, provide the accession numbers to the Brazilian samples of this study in phylogenetic trees.

2. The authors have not provided any figures to substantiate clinical signs and gross findings.

3. Do the polymorphisms mentioned in Table 2 influence the amino acid substitutions? Also, not clear about the amino acid gap mentioned. I was wondering about the possibility of translating the sequences and comparing.

4. The authors did not investigate the transmission route, specifically in this study. Therefore, the transmission route description in the abstract is a bit overstated. May be fine with discussion. I was wondering about other transmission possibilities.

5. This manuscript needs English editing.

Minor points:

Line 147: 1179 bp?

Comments on the Quality of English Language

This manuscript would benefit from the editing of a native English speaker.

Author Response

Dear Editors and Reviewers:
We thank the reviewers for their thoughtful comments and constructive suggestions concerning our manuscript entitled “Molecular characterization of infectious laryngotracheitis virus (ILTV) involved in poultry outbreaks reveals virus origin and estimated spreading route” (viruses-3388251), which enabled us to resubmit a clearly improved manuscript. We highlighted the amendments in the revised manuscript, and responded, point by point to, the comments as listed below.

Reviewer #2: 
The authors aimed to describe a case of ILTV in Broiler and broiler breeder flocks located in the Northern and Western regions of Parana State of Brazil in the second semester of 2024. Then, genetically characterized based on the amplification of the ICP4 gene in two fragments, and phylogenetically analysed. Finally, possible transmission routes discussed.
General points:
Strength: Science is OK.
ILTV genotypes in broilers and breeder broilers in Brazil will add circulating genotypes in Brazil and would add information to the ILTV genotypes literature.
Weakness: One major issue is the lack of novelty of the work and limited interest in Brazil. Moreover, results analysis is just based on 164 ICP4 sequences, and other written results are descriptive. A complete genome analysis may give more information on phylogeny and the presence of any recombination events.
R0. We greatly appreciate the comments and suggestions on our manuscript.

Q1. Figure 1: The labeling of sequence names on the phylogenetic tree is inconsistent. For example, VI-1 is mentioned with the word “ILTV,” and for other sequence names, it is removed. May be just use the accession number with year and country, and specific names of the samples. In addition, provide the accession numbers to the Brazilian samples of this study in phylogenetic trees.
R1. We thank the reviewer for this observation. The labels literally correspond to what is designated as strain names in GenBank. Countries are located on the outside world. Now we have added the years of isolation of each strain as suggested.

Q2. The authors have not provided any figures to substantiate clinical signs and gross findings.
R2. We thank the reviewer for this observation. We have now added section 3.1. Gross findings in the results and photos (Figure S1) as suggested. The changes are highlighted in yellow.

Q3. Do the polymorphisms mentioned in Table 2 influence the amino acid substitutions? Also, not clear about the amino acid gap mentioned. I was wondering about the possibility of translating the sequences and comparing.
R3. We thank the reviewer for this observation. We have identified polymorphic sites in nucleotides that constitute changes in amino acids (symbol † in the table). We have also added a sentence to relate these changes. The modifications are highlighted in yellow.

Q4. The authors did not investigate the transmission route, specifically in this study. Therefore, the transmission route description in the abstract is a bit overstated. May be fine with discussion. I was wondering about other transmission possibilities.
R4. We thank the reviewer for this observation. We have modified the sentences in the abstract as suggested. Additionally, we have added a few sentences at the end of the discussion highlighting the need for complete genome studies in the region, which to date are non-existent in Brazil and make it impossible to make comparisons at the moment.

Q5. This manuscript needs English editing.
R5. We thank the reviewer for this observation. We have sent the corrected version of the manuscript for language review as suggested. Changes are highlighted in yellow.

Q6. Line 147: 1179 bp?
R6. We thank the reviewer for this observation. We apologize for that mistyping. We have corrected the error and added the meaning in full. The changes are highlighted in yellow.

Round 2

Reviewer 2 Report

Comments and Suggestions for Authors

Authors addressed all of my comments, and well appreciated.